# Non-Covalent BTK Inhibitors—The New BTKids on the Block for B-Cell Malignancies

**DOI:** 10.3390/jpm11080764

**Published:** 2021-08-03

**Authors:** Katharine L Lewis, Chan Y Cheah

**Affiliations:** 1Department of Haematology, Sir Charles Gairdner Hospital, Nedlands, WA 6009, Australia; katharine.lewis@health.wa.gov.au; 2Linear Clinical Research, Nedlands, WA 6009, Australia; 3Medical School, University of Western Australia, Crawley, WA 6009, Australia; 4Department of Haematology Pathwest, Nedlands, WA 6009, Australia

**Keywords:** Bruton’s tyrosine kinase, non-covalent, B-cell malignancies, B-cell receptor, BTK inhibitors

## Abstract

The B-cell receptor signalling pathway plays a critical role in development of B-cell malignancies, and the central role of Bruton’s tyrosine kinase (BTK) activation in this pathway provides compelling rationale for BTK inhibition as a therapeutic strategy for these conditions. Covalent BTK inhibitors (BTKi) have transformed the treatment landscape of B-cell malignancies, but adverse events and treatment resistance have emerged as therapeutic challenges, with the majority of patients eventually discontinuing treatment due to toxicity or disease progression. Non-covalent BTKi have alternative mechanisms of binding to BTK than covalent BTKi, and therefore offer a therapeutic alternative for patients with B-cell malignancies, including those who have been intolerant to, or experienced disease progression during treatment with a covalent BTKi. Here, we summarise the clinical data, adverse events and mechanisms of resistance observed with covalent BTKi and describe the emerging data for non-covalent BTKi as a novel treatment for B-cell malignancies.

## 1. Introduction

B cell receptor (BCR) signalling (see Figure 1) plays a key role in the development and functioning of adaptive immunity, especially B cell development and maturation. This signalling cascade involves many kinases, including Bruton’s tyrosine kinase (BTK). BTK is a member of the TEC family of kinases; a group of non-receptor protein tyrosine kinases characterised by a pleckstrin homology (PH) domain in the protein structure. The PH domain facilitates binding of phosphoinositides, allowing TEC family kinases to play a role in phosphotyrosine-mediated and phospholipid signalling. BTK was identified in 1993 as a key mediator in the BCR signalling process [1,2], with contributory roles in the innate immune system function including dendritic cells and macrophages. The BCR signalling pathway also plays a crucial role in the pathogenesis of B-cell malignancies including chronic lymphocytic leukaemia (CLL), Waldenstrom’s macroglobulinemia (WM), mantle cell lymphoma (MCL) and marginal zone lymphoma (MZL), driving abnormal proliferation and survival [3,4].

Antigen binding to the BCR triggers phosphorylation of CD79a and CD79b (a disulphide-linked IgA-IgB heterodimer), in turn leading to phosphorylation and activation of BTK by spleen tyrosine kinase (SYK) [3,5]. There are five protein binding domains on BTK, with two key tyrosine phosphorylation sites–Y223 in the SRC homology (SH3) domain and Y551 in the kinase domain. During BCR signalling, phosphorylation by SYK at Y551 initiates autophosphorylation of Y223 [6]. This activation of BTK initiates further complex downstream signalling involving the nuclear factor kappa-B (NF-KB) pathway, leading to nuclear translocation of NF-KB transcription factors [7].

The central role of BTK activation in BCR signalling provides strong biological rationale for BTK inhibition as an effective therapeutic intervention in B-cell malignancies, due to the resultant suppression of BCR signalling, and subsequent reduced B-cell proliferation and survival.

Following the discovery of BTK, numerous covalent BTK inhibitors (BTKi) have been developed to target BTK for application in the treatment of B-cell malignancies [8,9,10,11]. These oral agents have notable and durable efficacy across a range of B-cell malignancies including CLL, WM, MCL and MZL, and have transformed the treatment of these conditions. Several covalent BTKi including ibrutinib, zanubrutinib and acalabrutinib are FDA-approved in relapsed/refractory and/or treatment naïve settings in various B-cell lymphoma subtypes. However, most patients receiving covalent BTKi eventually experience treatment failure due the development of resistance or intolerance. Non-covalent BTKi have key differences in their structure and mechanism of action, offering important potential strategies to improve patient outcomes due to improved tolerability and retaining efficacy when covalent BTKi have failed.

## 2. Covalent BTKi

Several covalent BTKi have been shown to be highly effective for treatment of B-cell malignancies [8,9,10,11,12]. Ibrutinib, acalabrutinib and zanubrutinib all have FDA approval in this sphere, with indications across a range of B-cell malignancies in both treatment naïve and relapsed/refractory settings.

Ibrutinib (PCI-32765) was the first in class covalent BTKi developed for treatment of B-cell malignancies. It was initially discovered in 2006; selected during the development of a series of selective, irreversible BTKi for preclinical development in rheumatoid arthritis [13]. Preclinical efficacy in B-cell malignancies was first reported in 2010 [14] and results from a phase 1 clinical trial using ibrutinib to treat patients with B-cell malignancies was published in 2012 [15]. Ibrutinib is an irreversible, small molecule BTK inhibitor that asserts its action by binding covalently to the cysteine-481 binding site of BTK, blocking phosphorylation and subsequent activation, and suppressing downstream signaling [14]. There are now extensive trial and retrospective data demonstrating the effectiveness of ibrutinib in CLL, WM, MCL and other B-cell malignancies [8,10,16,17,18,19,20,21,22,23]. Ibrutinib results in frequent and durable responses including in patients harboring high risk biological features such as *TP53* aberrations [8,10,16,17,18,19,20,21,22]. When administered as frontline therapy to patients with CLL in the E1912 study [18], ibrutinib overcame the well documented poor prognosis consistently observed in patients with *TP53* aberrations in CLL.

Ibrutinib has also been combined with other agents in B-cell malignancies with good effect, including venetoclax and rituximab [24,25,26,27,28]. Preclinical studies suggest synergism of BTKi in combination with these agents, and clinical studies have demonstrated the ability of these combinations to produce deeper and more durable responses, including in patients with *TP53* disruptions [24,26,28].

Ibrutinib paved the way for the development of other covalent BTKis. Acalabrutinib (ACP-196) is a second-generation BTKi developed and evaluated using a canine model of B-non Hodgkin lymphoma [29]. Zanubrutinib (BGB-3111) is another second-generation BTKi, designed using structure-activity relationship drug design, and selected following in vitro potency studies, selectivity, pharmacokinetics and pharmacodynamic studies [30]. In recent years, acalabrutinib and zanubrutinib have also received FDA approval. These agents have been developed with enhanced selectivity to reduce ‘off-target’ (non-BTK) kinase inhibition. In randomized clinical studies this has translated to more favourable safety profiles with lower rates of atrial fibrillation, bruising, rash and diarrhea than with ibrutinib [10,12,31,32,33], Several other covalent BTKi remain in earlier clinical development, including TG-1701 [34], orelabrutinib [11] and tirobrutinib [35,36].

Despite the remarkable efficacy of covalent BTKi, treatment failure is frequently observed, due to either treatment resistance (primary or acquired) or intolerance, with cumulative discontinuation rates as high as 40% after 4 years of therapy [37]. Overcoming the mechanisms of resistance and reducing the frequency of severe or persistent adverse events are vital to improving patient outcomes.

### 2.1. Adverse Events with Covalent BTKi

BTKi treatment related adverse events are frequent. A number of underlying mechanisms have been established, including ‘off target’ inhibition of other kinases containing a BTKi-binding cysteine, binding to ERBB2/HER2 and ERBB4/HER4, and EGFR inhibition. In their pooled analysis of three randomized studies (RESONATE [22], RESONATE-2 [38], PCYC-1102/1103 [39]) where a total of 424 CLL patients were treated with ibrutinib Coutre et al. reported that 13% patients had required a dose reduction in ibrutinib due to adverse event, and a further 11%permanently discontinued [37]. Median follow up for the integrated analysis of RESONATE and RESONATE2 was 29 months (range 0.2–42.9), and for the PCYC-1102/1103 long term safety analysis was 47.9 months (range 0.3–67.4). The majority of adverse events were grade 1/2. The most frequently observed adverse events were diarrhea (52%; 5% grade 3), fatigue (36%; 3% grade 3). The most frequently reported grade 3/4 adverse events were neutropenia (18%) and pneumonia (12%). Other infection, bleeding, atrial fibrillation, anemia and hypertension were also observed.

Two randomized phase 3 studies comparing second generation BTKi with ibrutinib have recently been reported. In the ALPINE study,415 patients were randomized 1:1 to either ibrutinib or zanubrutinib, with a median follow up of 15 months [32]. In the ELEVATE RR study 533 patients were randomized 1:1 to either ibrutinib or acalabrutinib, with a median follow up of 40.8 months [33]. In ELEVATE RR, acalabrutinib was non-inferior to ibrutinib in terms of efficacy, with lower rates of atrial fibrillation (9.4% vs. 16.0%), hypertension (9.4% vs. 23.2%), bleeding (38.0% vs. 51.3%) and lower rates of adverse events leading to discontinuation (14.7% vs. 21.3%). At a preliminary response assessment in the ALPINE study, zanubrutinib had superior response rates to ibrutinib (78.3% vs. 62.5%), with lower rates of atrial fibrillation (2.5% vs. 10.1%), major bleeding (2.9% vs. 3.9%) and adverse events leading to discontinuation (7.8% vs. 13.0%). Rates of grade ≥3 infections were also reduced in patients treated with zanubrutinib compared with ibrutinib (12.7% vs. 17.9%), despite higher rates of neutropenia (28.4% vs. 21.7%).

In a recent analysis of the mechanisms underlying adverse events during BTKi use [40], the authors summarized clinical trial and real world adverse event data for covalent BTKi, reporting overall discontinuation rates due to adverse events of 9–23% in clinical trials [9,10,22,41,42,43] and 23–49% in the community setting [43,44].

### 2.2. Mechanisms of Resistance with Covalent BTKi

Despite their efficacy, resistance to covalent BTKi is frequently observed. Ibrutinib resistance was initially observed in patients who experienced progression while receiving ibrutinib in clinical trials, and mechanisms of resistance identifed through subsequent laboratory testing [45]. Primary BTKi resistance in CLL is rare and the mechanisms remain poorly understood [46,47,48] but is more frequent in MCL [49], observed in around 30% of patients. However, over time many patients with CLL and other B-cell malignancies develop secondary resistance to BTKi therapy. Although a number of different resistance mechanisms have been described, with variation across disease subtypes, the problem remains incompletely understood. In CLL, the most frequent mechanisms of resistance observed in ibrutinib treated patients are point mutations: the BTK mutation C481S (a mutation in BTK altering the configuration of the cysteine binding site) and mutations in phospholipase C gamma 2 (PCLg2) [50]. These two mutations are collectively observed in up to 80% patients with CLL and progression on ibrutinib therapy [51,52]. C481S mutations have also been observed in up to 40% patients who develop Richter’s transformation on ibrutinib [52,53] and in ibrutinib resistant WM [54]. They have also been observed, but less frequently, in patients experiencing progression of MCL during BTKi therapy [5]. C481 mutations have also been described in acalabrutinib [55] and zanubrutinib [56] treated patients with CLL experiencing disease progression. In zanubrutinib treated patients, concurrent BTK C481 mutations and BTK Leu528Trp mutations have been reported [56]. Other resistance mechanisms to covalent BTKi described in CLL and B-cell lymphomas include genomic and epigenetic activation of downstream signalling pathways [57]. Target inhibition may be incomplete in patients with highly proliferative malignancies treated with covalent BTKi, due to the low oral bioavailability, short half-life and high protein binding of these agents, potentially further contributing to, or driving, drug resistance.

## 3. Non-Covalent BTKi—Clinical Development

Non-covalent BTKi exert their inhibition of BTK by different mechanisms to covalent BTKi. They do not act by binding to the C481 site on BTK, and therefore offer a potential alternative therapeutic option to patients with B-cell malignancies than covalent BTKi, including those who have developed acquired resistance due to BTK C481 mutations following prior therapy with a covalent BTKi. To our knowledge there are four non-covalent BTKi that have reached clinical development. Key characteristics of each are summarized in Table 1, along with details on the status of clinical development. MK1026 (formerly known as ARQ-531) binds to BTK by forming hydrogen bonds with E475 and Y476; fenebrutinib forms hydrogen bonds with K430, M477 and D539; pirtobrutinib (LOXO-305) blocks the ATP binding site of BTK.

### 3.1. Pirtobrutinib (LOXO-305)

Pirtobrutinib is an oral, highly selective, reversible BTKi with nanomolar potency against both wild-type and C481-mutated BTK [58]. Pirtobrutinib is highly selective on BTK, with >300 fold selectivity on BTK over 98% of other kinases, reducing the potential for ‘off-target’ toxicities. It has been designed to maintain greater than 90% of maximal BTK inhibition at trough, thereby achieving effective target inhibition throughout the dosing interval, even in proliferative tumours.

Results from the first-in-human, phase 1/2 clinical trial (BRUIN) were recently reported [59]. Eligible patients had B-cell malignancies and had received at least two prior lines of therapy, or with one prior line of therapy if a covalent BTKi had been received first line. BTKi exposed and naïve patients were included, and patients were eligible regardless of C481 mutation status. All patients received pirtobrutinib once daily as monotherapy, continuously in 28-day cycles until unacceptable toxicity or disease progression. The maximum tolerated dose was not reached during phase 1, and the recommended phase 2 dose was 200 mg once daily.

Three hundred and twenty-three patients (170 with CLL/small lymphocytic lymphoma (SLL), 61 with MCL, 26 with WM and 66 with other B-cell lymphomas) were treated. High risk biological features were prevalent, including 17p- in 25% and TP53 mutation in 30% of those with CLL. Eighty-six (86%) patients with CLL and 93% patients with MCL had received a prior covalent BTKi. Response rates were similar in patients who had discontinued prior covalent BTKi therapy regardless of whether it was due to progression or treatment toxicity. The overall response rate (ORR) was 63% in CLL/SLL (79% in those with 17p- or TP53 mutation), 52% in MCL, 68% in WM and 50% in follicular lymphoma (FL).

Of the efficacy evaluable patients with WM (*n* = 19), 9 experienced partial response and 4 experienced minor response. Ten (77%) of responding patients remained on treatment with median follow up of 5 months.

Of the efficacy evaluable patients with MCL (*n* = 56), 14 experienced complete response and 15 experienced partial response. Responses were observed in patients who had receive prior autologous and allogeneic stem cell transplant (ORR 64%; *n* = 9), and CAR T-cell therapy (ORR 100%, *n* = 2). Responses were observed early, with a median time to response of 1.8 months. Fifty seven percent of patients with MCL remained on pirtobrutinib at the time of reporting, with median follow up of 6 months.

Patients with “triple-refractory” CLL (covalent BTKi, PI3K inhibitor and venetoclax) had an ORR of 58%, encouraging results for a growing group of patients with unmet therapeutic need. Responses were also observed in 6/25 (24%) patients with diffuse large B-cell lymphoma and 2/9 (22%) patients with marginal zone lymphoma.

The most frequently reported adverse events were fatigue (20%), diarrhoea (17%), contusion (13%) and neutropenia (13%). Adverse events were predominantly grade 1/2, with the notable exceptions of neutropenia (grade 3; 6%, grade 4; 4%; all uncomplicated) and anaemia (grade 3; 4%). Adverse events commonly observed during covalent BTKi therapy were less frequent with pirtobrutinib therapy and almost exclusively grade 1/2 (bruising 16%, rash 11%, arthralgia 5%, hypertension 5% and atrial fibrillation/flutter 1%) [59].

Data from a subset of patients on the BRUIN study with previously treated Richter’s transformation (RT) (*n* = 17) were recently reported demonstrating encouraging activity 67%, CR 13%) and favourable toxicity profile (consistent with that observed in other disease subtypes on the BRUIN study), in this challenging disease [60]. This patient cohort was heavily pre-treated, with a median number of total prior lines of therapy (for CLL and RT) of 6 (range 2–10), and median prior lines of therapy for RT 2 (range 1–5). Fourteen patients (82%) had prior covalent BTKi exposure. However, the median follow up period was short at 3.7 months, and longer follow up is required to determine durability of these early favorable responses.

In summary, pirtobrutinib was well tolerated, and therapeutic activity was observed at all dose levels across a range of B-cell malignancies previously treated with covalent BTKi, including those with C481 mutations, uncharacterised resistance mechanisms and patients who had discontinued BTKi therapy previously due to treatment intolerance. It is worth noting that at the time of reporting, median follow up was around 6 months across disease subtypes and given the long natural history of many B-cell malignancies, longer follow up is required to determine the durability of these responses. However, the early efficacy and tolerability data for this agent is encouraging.

The phase I/II BRUIN trial is also exploring combination dosing regimens, where patients with CLL/SLL receive pirtobrutinib in combination with either venetoclax (an oral B-cell lymphoma 2 inhibitor (BCL2i) or venetoclax and rituximab. In both treatment combinations, venetoclax is introduced from cycle two to allow reduction in tumour bulk during the first treatment cycle, thereby reducing the risk of tumour lysis syndrome prior to initiating venetoclax; a well recognised and potentially life-threatening complication following initiation of BCL2i therapy. Results are awaited.

Phase 3, randomised trials involving pirtobrutinib are ongoing. In the BRUIN-MCL-321 study (NCT04662255), eligible patients with BTKi-naïve MCL, and at least one prior line of therapy are being randomised to receive either pirtobrutinib or investigator’s choice of available covalent commercially BTKi (ibrutinib, acalabrutinib or zanubrutinib). In the BRUIN CLL-321 study (NCT0466038), patients with CLL or SLL with prior BTKi exposure are randomised to receive pirtobrutinib or investigator’s choice of either idelalisib plus rituximab or bendamustine plus rituximab.

### 3.2. Fenebrutinib (GDC-0853)

Fenebrutinib is a selective, reversible, non-covalent BTKi. It exerts it inhibitory activity on BTK by forming hydrogen bonds with K430, M477 and D539, rather than by interaction with the C481 residue, therefore has potential for therapeutic activity in patients with C481 mutations. The first-in-human phase 1 study was reported in 2018, with 24 patients having been treated in a phase I dose escalation study [61]. Six patients had a documented C481S mutation at enrolment; 1/6 of these had a treatment response. The ORR was 33%, with median duration of response of 3.8 months. Response rates were superior in CLL (50%, *n* = 7), with only 10% of patients (*n* = 1) with other B-cell malignancies exhibiting a disease response-a patient with MCL who attained a durable complete remission [61]. Adverse events were common, but almost exclusively grade 1/2 and low discontinuation rates for toxicity. The most frequently observed events were fatigue (37.5%), nausea (33%), diarrhea (29%), thrombocytopenia (25%), headache (21%), dizziness (17%), abdominal pain (17%) and cough (17%). There were two fatal adverse events of H1N1 influenza and influenza infection. The study was terminated and development of this agent has been discontinued in B-cell malignancies, but continues in other disease including multiple sclerosis (NCT04544449).

### 3.3. Vecabrutinib (SNS-062)

Vecabrutinib is a selective, reversible, non-covalent BTKi with nanomolar potency. Preclinical studies confirm activity against wild-type and C481 mutated BTK [62]. It has activity against IL2-inducible T-cell kinase (ITK) but not epidermal growth factor receptor (EGFR), thereby reducing EGFR associated toxicities, such as diarrhea and rash, observed with ibrutinib and other covalent BTKi. A phase 1, first-in-human study was completed, where 32 healthy participants were treated with vecabrutinib [63]. Maximum tolerated dose was not reached and all adverse events were grade 1/2. Headache was the most frequent adverse event (16%), with nausea, constipation, bronchitis, fatigue, and supraventricular tachycardia also observed.

A phase Ib study has been completed in patients with B-cell malignancies (NCT03037645). Thirty-nine patients were treated in dose escalation until time of progression or unacceptable toxicity. Interim data for the first 27 patients were reported [64]. All patients had received a prior covalent BTKi, 48% had BTK C481 mutations and 20% had PCLG2 mutations. Next generation sequencing revealed a median of 5 mutations per patient, most commonly SF3B1 (24%), NOTCH1 (20%) and ATM (20%). Sustained inhibition of BTKi (range 48–100%) was observed at the end of the first cycle of treatment. The phase Ib trial was completed and the final results are awaited. However, while vecabrutinib was well tolerated with a favourable safety profile, there was insufficient clinical activity to proceed with the planned phase 2 portion of the study and development of this agent in B-cell malignancies has not been further progressed [65].

### 3.4. MK-1026

MK-1026 (formerly ARQ-531) is a potent, reversible inhibitor of wild-type and C481 mutant BTK, with nanomolar potency; greater for C481 mutant BTK (IC50 = 0.39 nM) than wild type (IC50 = 0.85 nM) [66]. It is less selective than the previously discussed non-covalent BTKi, with significant activity on other kinases including SRC, ERK and AKT. In vitro studies confirm inhibition of BCR signalling, via inhibition of BTK [66]. As well as inhibiting BTK, MK-1026 can also inhibit signalling downstream of PCLG2, including in cell lines with PCLG2 mutations; an alternative pathway to exert therapeutic action in B-cell malignancies [66]. A phase 1 dose-escalation study, in which 40 patients received MK-1026 to treat a range of relapsed/refractory B-cell malignancies, has been completed [67]. Most patients (26/40) had CLL (85% with BTK C481S mutation); other histologic subtypes included Richter’s transformation, DLBCL, FL, and MCL. Patients were heavily pre-treated, with a median of 4 prior lines of therapy, and all patients had received a prior covalent BTKi. The recommended phase 2 dose has been determined at 65 mg once daily. The most frequent adverse events reported were nausea (10%), diarrhoea (10%), fatigue (8%), neutropenia (8%), dysgeusia (8%) and rash (8%). While the majority of adverse events were grade 1/2, grade 3/4 adverse events were also observed including neutropenia (8%), and febrile neutropenia, cellulitis, thrombocytopenia, lipase elevation and rash (3% each). The ORR was 25%, and responses were observed across a range of histologic subtypes. The study is ongoing in phase Ib and a larger phase II study (MK-1026-003; NCT04728893) is currently ongoing with MK-1026 in patients B-cell malignancies, planned to enrol up to 400 patients.

## 4. Non-Covalent BTKi—Preclinical Development

As well as those described above with clinical trial data, there are a number of non-covalent BTKi in preclinical development. XMU-MP-3 [68], CB1763 (AS-1763) [69] and GNE-431 [70] all have in vitro activity against both wild type and C481 mutant BTK. Additionally, XMU-MP-3 and CB1763 have significant antitumour activity in vivo in xenograft models [68,69]. Another agent CGI-1746 inhibits BTK by stabilising the inactive form [71], but no publically available in vivo data are available at the time of writing. Clinical trials are required to investigate safety and efficacy of these agents further.

## 5. Combinations

Synergism between covalent BTKi and other agents has been discussed previously. As well as pirtobrutinib monotherapy, the BRUIN trial offers eligible patients with CLL and small lymphocytic lymphoma (SLL) treatment with combination of pirtobrutinib and venetoclax ± rituximab. Venetoclax and rituximab have been safely administered in combination with other BTKi as previously discussed, and based on the favourable toxicity profile of pirtobrutinib monotherapy [59], the combination with pirtobrutinib is anticipated to be well tolerated. Data from these arms are awaited.

## 6. Conclusions

Covalent BTKi have improved outcomes for patients with B-cell malignancies but they are non-curative. There are some therapeutic options for patients refractory to, or intolerant of, covalent BTKi therapy, including venetoclax and PI3K inhibitors. Nonetheless, there remains unmet therapeutic need in the treatment of B-cell malignancies, especially for treatment of the increasing group of patients refractory to BCL2 inhibition, PI3k inhibition and BTK inhibition—‘triple refractory’ patients. Non-covalent BTKi may offer a potential therapeutic approach. Several agents exhibit a favourable safety profile and encouraging efficacy across a variety of B-cell malignancies, but the durability of response remains unanswered at present due to relatively short follow up. There is compelling biological rationale for combining non-covalent BTKi with other novel agents to induce deeper and more durable responses, and data from covalent BTKi combination regimens to support this approach. We anticipate the results of randomized trials of these agents will lead to the availability of more effective and well tolerated agents for patients with B-cell malignancies who have experienced covalent BTKi treatment failure. 

## Figures and Tables

**Figure 1 jpm-11-00764-f001:**
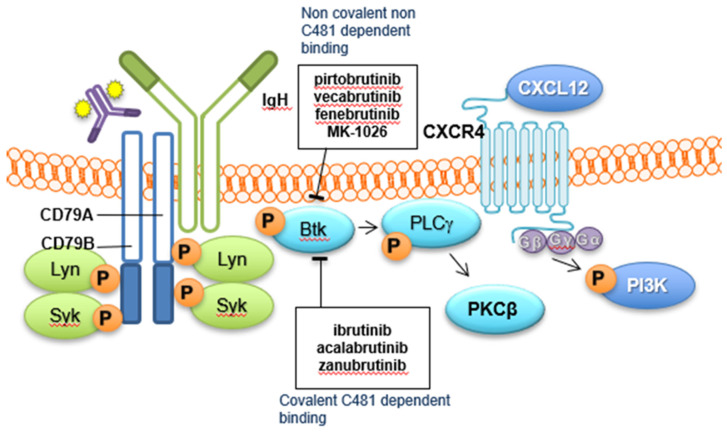
B-cell receptor (BCR) complex-covalent and non-covalent BTKi mechanism of action.

**Table 1 jpm-11-00764-t001:** Summary of characteristics of non-covalent BTKi in clinical development.

Non-Covalent BTKi	Pirtobrutinib	Fenebrutinib	Vecabrutinib	MK-1026
Structure	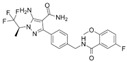	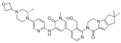	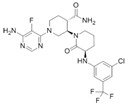	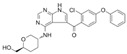
Other names	LOXO-305	GDC-0853	SNS-062	ARQ-531
Binding to BTK	Blocks ATP binding site of BTK	Hydrogen bonds with K430, M477, D539	Decreases surface expression of B-cell activation markers	Hydrogen bonds with E475, Y476
Other enzyme activity	Minimal	Minimal	Activity on ITK No activity on EGFR	Activity on SRC, ERK, AKT Inhibits signalling downstream of PCLG2
Side effects (%)	Fatigue (20%) Diarrhea (17%) Contusion (13%) Neutropenia (13%)	Fatigue (37.5%) Nausea (33%) Diarrhea (29%) Thrombocytopenia (25%) Headache (21%)	Anemia (37.5%) Headache (25%) Neutropenia (25%) Night sweats (25%)	Nausea (10%) Diarrhea (10%) Fatigue (8%) Neutropenia (8%) Dysgeusia (8%) Rash (8%)
Clinical development	Phase Ib/II ongoing in B-cell malignancies Phase III ongoing in MCL Phase III ongoing in CLL	Terminated in B-cell malignancies Ongoing in multiple sclerosis	Terminated in B-cell malignancies	Phase Ib ongoing in B-cell malignancies Phase II ongoing in B-cell malignancies
Key publications	[51,52]	[53]	[5,54,55]	[57,58]

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
