# Peer review of "Non-Covalent BTK Inhibitors—The New BTKids on the Block for B-Cell Malignancies"

_jpm, 2021, doi:10.3390/jpm11080764_

Round 1

Reviewer 1 Report

  1. The authors should refrain from using abbreviations in abstract. It is better to use expanded or full name of BCR and BTK. The first line of introduction also needs the expanded forms and after that authors can use the abbreviations. 
  2. The authors should mention tec family of protein tyrosine kinases in line 29 of introduction.
  3. In figure 1, authors should consider inserting downstream effects of PI3K activation or PKCbeta activation such as cytoskeletal remodeling or cell proliferation/survival/migration.
  4. Before line 46, authors should consider introducing covalent and noncovalent inhibitors (history of their discovery) and caveats of using them. 
  5. Was the resistance to covalent BTKi seen during the clinical trials or laboratory experiments? The authors should discuss this. 
  6. Table X is not shown in the manuscript.
  7. History of how each BTKi inhibitor was discovered needs to be added in the text (at least for PCI- 32765 and each of the non covalent inhibitors).
  8. A cartoon in the end showing how different covalent and noncovalent inhibitors work on BTK kinase will be useful. The authors can also add specificity or other relevant information (clinical trial stage etc) for readers to get a quick graphical overview of the review. 

Author Response

1. The authors should refrain from using abbreviations in abstract. It is better to use expanded or full name of BCR and BTK. The first line of introduction also needs the expanded forms and after that authors can use the abbreviations.

Thank you for identifying this.  The abstract has been revised to define all terms without abbreviation before they are used.  Terms are also used again in full at the time of first use in the main review article.  Amendments are highlighted in red.

2. The authors should mention tec family of protein tyrosine kinases in line 29 of introduction.

A definition and brief explanation of the Tec family of protein kinases has been added in Section 1, line 31-34 and is highlighted in red.

3. In figure 1, authors should consider inserting downstream effects of PI3K activation or PKCbeta activation such as cytoskeletal remodeling or cell proliferation/survival/migration. Thank you for this suggestion.  We have chosen to focus the figure on the pathways more immediately surrounding BTK.

4. Before line 46, authors should consider introducing covalent and noncovalent inhibitors (history of their discovery) and caveats of using them.

Thank you for this suggestion. A brief explanation of discovery each covalent BTKi has been inserted into the relevant sections of the review (section 2, lines 65-68 and 82-87).  Additional relevant references have been included. 

5. Was the resistance to covalent BTKi seen during the clinical trials or laboratory experiments? The authors should discuss this. 

Thank you for the comment. The identification of ibrutinib resistance was first observed in patients progressing on ibrutinib in clinical trials and then reverse translated at the bench. This has been briefly discussed in section 2, lines 129-131.

 6. Table X is not shown in the manuscript.

Thank you for identifying this error– this was a historical addition left in place in error from an earlier iteration of the manuscript.  All references to 'Table X’ have now been removed.

7. History of how each BTKi inhibitor was discovered needs to be added in the text (at least for PCI- 32765 and each of the non covalent inhibitors).

Thank you for this suggestion. As for comment 4, a brief explanation of the discovery of ibrutinib, acalabrutinib and zanubrutinib has been inserted into the relevant article sections (section 2, lines 65-68 and 82-87).

Reviewer 2 Report

The authors review BTK inhibitors used to treat B cell malignancies. Several of the inhibitors are considered as potential medicines and are evaluated via clinical trials. 

The manuscript is well-organized. There are multiple typos, however, that need to be fixed. E.g., line 78 "Table X" and line 79 "inhibitionIn", etc.

A large section of the review lacks proper references. E.g., Lines 166 to 220.

The authors list a number of companies in the COI section. Is it possible to specify whether COI involves BTKi or not?

Author Response

  1. The manuscript is well-organized. There are multiple typos, however, that need to be fixed. E.g., line 78 "Table X" and line 79 "inhibitionIn", etc.

Thank you and apologies for the oversight – typos have been corrected.  ‘Table X’  was a historical addition left in place in error from an earlier iteration of the manuscript.  All references to 'Table X’ have now been removed.

  1. A large section of the review lacks proper references. E.g., Lines 166 to 220.-

 Thank you for the comment.  This section is exclusively referenced from the single publication of clinical trial data for pirtobrutinib, as already referenced.  The data is discussed in detail in this section, but there are no additional references to add.

  1. The authors list a number of companies in the COI section. Is it possible to specify whether COI involves BTKi or not?

Thank you; several of the companies listed have a BTK either commercially available of in development.  A sentence has been added within the COI section to declare this explicitly.